# Effect of Laser Parameters on Processing of Biodegradable Magnesium Alloy WE43 via Selective Laser Melting Method

**DOI:** 10.3390/ma13112623

**Published:** 2020-06-09

**Authors:** Jan Suchy, Miroslava Horynová, Lenka Klakurková, David Palousek, Daniel Koutny, Ladislav Celko

**Affiliations:** 1Faculty of Mechanical Engineering, Institute of Machine and Industrial Design, Brno University of Technology, Brno 60190, Czech Republic; David.Palousek@vut.cz (D.P.); Daniel.Koutny@vut.cz (D.K.); 2Central European Institute of Technology, Research Group of Materials Characterization and Advance Coatings, Brno University of Technology, Brno 60190, Czech Republic; Miroslava.Horynova@ceitec.vutbr.cz (M.H.); Lenka.Klakurkova@ceitec.vutbr.cz (L.K.); ladislav.celko@ceitec.vutbr.cz (L.C.)

**Keywords:** Mg-alloy, single track, additive manufacturing, thin wall, 3D printing, surface quality

## Abstract

The global aim of the theme of magnesium alloy processing by the selective laser melting technology is to enable printing of replacements into the human body. By combining the advantages of WE43 magnesium alloy and additive manufacturing, it is possible to print support structures that have very similar properties to human bones. However, printing magnesium alloy parts is very difficult, and the printing strategies are still under development. Knowledge of weld deposit behaviour is needed to design a complex printing strategy and still missing. The main aim of the manuscript is the find a stable process window and identify the dependence of the weld deposit shape and properties on the laser power and scanning speed. The range of the tested parameters was 100–400 W and 100–800 mm/s for laser power and scanning speed. The profilometry and light microscopy were used to verify the continuity and shape evaluation. The microhardness and EDX analysis were used for the detailed view of the weld deposit. The manuscript specifies the weld deposit dimensions, their changes depending on laser power and scanning speed, and the continuity of the weld tracks. The stable weld deposits are made by the energy density of 5.5–12 J/mm^2^. Thin walls were also created by layering welds to determine the surface roughness scattering (Ra 35–60) for various settings of laser power and scanning speed.

## 1. Introduction

The materials used for implant replacements are corrosion-resistant steels, titanium and alloys based on cobalt and chromium. The disadvantage of these biomaterials is the potential contamination of tissue by metal particles as a result of corrosion and/or abrasion [1,2], which may lead to a decrease in biocompatibility and loss of tissue [3,4,5]. Moreover, the elastic modulus of these biomaterials does not match that of bone tissues, which leads to stress effects that may reduce stimulation of new bone growth and bone repair, reducing implant stability [6]. Furthermore, when these materials are used for implants, the patient usually has to undergo reoperation to remove screws, plates and pins used to stabilise the fracture.

Magnesium is one of the basic building blocks of the human body and is naturally found in human bone [7,8,9]. Its corrosion leads to the creation of a nontoxic oxide that is then expelled from the body through urine [10]. In addition, its presence in bones contributes to stimulating new bone tissue growth [11,12,13,14]. This, combined with magnesium’s elastic modulus, which is closer to that of bone tissues than other biomaterials (bone: 3–20 GPa, Mg: 41–45 GPa, Ti alloy: 110–117 GPa, corrosion-resistant steels: 189–205 GPa) [10] delays or, in some cases, completely removes the need for implant reoperation. Tests on animal tissues have proven that a magnesium implant nourishes injured tissue for 12–18 weeks and helps maintain its mechanical integrity before it is replaced by natural tissue [14,15]. WE43 stands out among magnesium alloys due to its corrosion resistance [16,17]. It also shows no signs of mutagenicity during in vitro testing [18]. These properties make it a suitable candidate for biomedicinal use.

Magnesium and its alloys have low density, good ductility, better shock-absorption properties than aluminium, excellent castability and good weldability. They also have their downsides, however, including lower fatigue strength than aluminium, high reactivity and very low corrosion resistance [19]. In addition, magnesium alloys are very reflective (up to 98%), have a low melting point (650 °C) and evaporation point (1091 °C), and low density (1738 kg/m^3^). These properties, along with susceptibility to corrosion, are the main reasons why problems are encountered in processing magnesium and its alloys through 3D printing. 

At the same time, processing magnesium alloys via metal 3D printing could lead to the production of porous lattices, whose geometry is controlled by CAD data and which, otherwise, are not producible by conventional means [19]. Selective laser melting (SLM) appears to be a suitable technology for processing magnesium, as it is well suited for processing different material classes, such as Al, Cu, CoCr, Ti, Au, Ni, etc. [20,21].

Several publications have already examined the processing of magnesium alloys via SLM [22,23,24,25,26,27,28,29]. However, these studies have mainly focused on magnesium alloys with aluminium (AZ91, AZ31) or pure Mg. 

Nevertheless, several research teams have studied the processing of WE43 alloy via SLM in particular [30,31,32,33,34,35,36,37,38]. Results show that the material can be processed using SLM technology to achieve a relative density of over 99%. However, the individual studies were conducted using different machines and different process parameters. The reasons for these different approaches are not explained in detail in these studies and information on the effects of individual process parameters on the production process of WE43 is missing. This creates a need for discussion regarding the manufacturing strategy and its improvement. Samples with lower surface quality are often referred to as a deficiency of SLM processing of Mg alloys. This deficiency can be resolved by a detailed weld deposit analysis and the adjustment of the weld deposit overlap and input energy density, as with other materials [39,40,41]. The aim of this study is to clarify the effects of laser power and scan speed on the production process, to identify the dependence of size and continuity of weld tracks on energy density and to perform a basic analysis of surface quality, depending on the process parameters.

## 2. Materials and Methods 

### 2.1. Sample Production

The samples used for testing were weld tracks and walls with a thickness of one weld deposition. The samples were produced using a continuous Ng-YAG laser, with a maximum power of 400 W (SLM 280^HL^, SLM Solutions Group AG, Lübeck, Germany). The laser focus was 82 μm, with a Gaussian profile and wavelength of 1060 nm. The samples were produced on magnesium base plate. The plate was preheated to 120 °C. The input material used was atomized powder, produced by Magnesium Elektron UK, with particle size distribution of 28–60 μm (Figure 1). Particle size distribution guaranteed by the manufacturer is D10–26.9 μm; D5–39.8 μm; D90–57.9 μm. The listed values were verified using a laser diffraction analyser (LA-960, Horiba, Kyoto, Japan). The chemical composition of the powder was in line with the standard ASTM B93/B93M-06 (Table 1). The surface of the particles was sufficiently covered by oxides Y2O3 and MgO (LYRA3, XMH, SEM, TESCAN, Brno, Czech Republic). SiO2 also appeared locally on the particle surface. The chosen thickness of applied layer (Lt) was 50 μm due to powder distribution typical for this layer thickness. The powder humidity before application of the first layer was 3.6% (Humidity-temperature probe, B + B Sensors). The oxygen concentration during construction was kept at under 0.1% O2 using a constant supply of compressed argon.

### 2.2. Analysis of Shape and Continuity

The aim of the test was to identify combinations of laser power (Lp) and scan speed (Ls) that lead to a stable melting process. Analysis of the shape and continuity of the weld tracks, produced through various combinations of Lp and Ls, was used to provide basic insight into the production process. Surface energy density, which is used in the study as a comparative criterion, was calculated using Equation (1). Weld depositions for the chosen combinations were made in two directions: in the direction of the flow of inert atmosphere, and against it. The focus was on differences in the track shape. The tested combinations were within the Lp and Ls ranges of 100–400 W and 100–750 mm/s, respectively. The dimensions of the individual weld tracks were determined based on metallographic sections or weld track scans (Figure 2). The weld track scans were made using profiler Contour GT-X (Bruker Corporation, Billerica, MA, USA). The scans were then used to evaluate weld track continuity, height and width. The values stated in the study were calculated, as an average, from ten measurements. The readability of the profilometric scan was 3 μm. The weld penetration was determined based on metallographic sections perpendicular to the weld track. Metallographic sections of all samples were prepared via wet grinding and polishing using diamond pastes. The macrostructure of the weld depositions was developed by chemical etching using Nital etchant. Observation and documentation of all metallographic samples was performed using a DSX510 opto-digital microscope (Olympus, Tokyo, Japan), which uses light field microscopy and/or DIC.
E = Lp/(Ls⋅Lt) (1)
where Lp is laser power; Ls is scan speed; Lt is layer thickness.

The test was then repeated with thin-wall samples, but only for Lp and Ls combinations chosen based on the results of the weld deposition testing to reduce the number of samples. The purpose of thin-wall testing was to compare the wall width to the weld deposition width, and to get closer to real weld deposition thickness found in volumetric specimen printing. Each thin-wall sample was created by layering weld tracks on top of each other. The shape of the produced walls corresponded to rectangular contours with floorplan dimensions of 10 mm × 2 mm; with one side of the rectangle produced in the direction of the flow of inert atmosphere and the other against it. The wall height was 10 mm, including a 2 mm block support. The wall thickness was determined based on a metallographic section, perpendicular to the weld track (similar to weld depositions). The wall thickness value was determined based on an arithmetic average from ten points (measurements). Thin-wall samples were observed and documented in an unetched state. The evaluation of samples was done via image analysis (Vision64®-64bit 5.60, Bruker Corporation, Billerica, MA, USA). The measurement error was determined based on image analysis readability and had a value of 3 μm.

### 2.3. Analysis of Surface Quality

Thin-wall samples were tested at various Lp and Ls combinations, and a qualitative measurement of surface quality was taken using a Contour GT-X profilometer (Bruker Corporation). The aim was to assess the effects of the process parameters on changes in surface quality. The measured area was located on the side of the thin-wall sample and was sized 1 mm × 9 mm. The surface of the sample was not treated before scanning. When performing the analysis, slant and wave of the sample surface was eliminated using Gaussian filters.

### 2.4. Measurement of Microhardness and Local Chemical Composition

To classify a stable combination of process parameters, microhardness was measured in the weld deposition cross-sections, in accordance with Vickers HV0.01. The aim was to uncover any potential dependence of changes in weld deposition hardness on the selected process parameters. Measurements were taken in two independent series. One was taken within the axis of the weld track, perpendicular to the base plate, and the other parallel to base plate, below the sample surface. Measurements were always taken on two weld depositions per combination of process parameters: one in the direction of flow of inert atmosphere, and the other against the flow. Hardness was measured using a Duramin-100 AC3 (Struers, Detroit, Cleveland, OH, USA) fully automatic hardness tester with a wide load range. 

Change in the chemical composition of the deposition material as a result of evaporation during fusion, mixing of plate material with weld deposition material or the formation of new phase structures were analysed, using local chemical EDX microanalysis with the help of an energy dispersive analyser (Oxford Instruments, Oxford, UK), an accessory of the TESCAN LYRA3 XMH scanning electron microscope. Microanalysis of the chemical composition was performed with the following parameter settings: the beam intensity index 14; the high voltage 20 kV. Under these conditions, the measurement error was within ±0.5 at.% depending on the atomic number of the analyzed element.

## 3. Results

### 3.1. Analysis of Shape and Continuity

The continuity of the weld tracks was evaluated based on the surface scanned by the profilometer. Combining different Lp and Ls values resulted in different weld deposition shapes. No significant differences were found when testing weld depositions in and against the direction of flow of the inert atmosphere.

The range of the tested Lp and Ls values was 100–400 W and 100–800 mm/s, respectively, in 25 W and 50 mm/s steps. The results can be divided into three categories, defined by the amount of surface energy introduced, since surface energy was used as a comparative criterion (Figure 3). Since the energy of individual areas partially overlapped, results were further divided based on the Ls and Lp combinations used. Weld tracks in the first area (2.5–8 J/mm^2^) were produced using insufficient energy. The powder material only partially melted under the effects of the laser and fusion with the base plate was insufficient. Many cases showed evidence of a balling effect. The result was a discontinuous weld track. The second area (13–80 J/mm^2^) was characterized by wide weld tracks produced through high-energy input. The melting pool was wide and lacked stability. High local energy resulted in burning of the material and the production of fumes (dense black mist). The last area (7–13 J/mm^2^) included visually continuous weld depositions with stabilised shapes. The dimensions of weld depositions in this area showed lower dispersion of values.

This initial testing was followed by an analysis of metallographic sections. The width and depth of weld depositions decreased as the energy density input decreased. The parameter with the largest influence on the weld deposition shape was Lp (Figure 4). The edges of areas delineated in the process map (Figure 3), based on continuity testing, corresponded to the results obtained from weld track sections. The insufficient energy area was characterised by narrower weld tracks with low depth. Combinations of weld depositions in the unstable area (Figure 3) were sensitive to any changes in Lp and Ls. The area contained weld depositions with deep penetration but also ones with insufficient energy. The melting pool of the depositions splashed, and many unmelted powder particles and drops of hardened melt were fused onto their surface (Figure 5b). The increase in weld deposition height, in the unstable area, was in the order of units of micrometres. The high energy density area was characterised by deep weld depositions. At times, their depth reached values over 600 μm. The weld tracks on samples from the high energy area were wider. The weld depositions made at Ls to up to 125 mm/s often contained pores in the weld root (Figure 5a), which reached sizes of 50–75 μm at a laser power of over 300 W. Small pores appeared in other weld depositions as well, regardless of the surface energy density. The spherical shape of the pores indicates trapping of fume bubbles in the rapidly hardening deposition material. Since initial tests have shown that Lp and Ls combinations of 300–400 W and 100–400 mm/s create tracks that are too deep and discontinuous, this area was not studied in further detail. The prospective area contained continuous weld depositions with a width of 200–250 μm, depth of 200–300 μm and a height of 20–40 μm.

As the height of a thin-wall sample grows and layers of already hardened powder are constantly remelted, thermal energy accumulates in the sample. That leads to a change in the dimensions of the sample. To quantify these changes, a thin-wall test was prepared, allowing the weld track width to be monitored under more real conditions. Similar to the weld deposition test, Lp had a dominant influence on the resulting shape of thin-walls. The surface of the thin walls was, however, very rough due to the fused particles of the atomised powder. This led to a high dispersion of values when measuring the width of the thin walls. Despite this, measurements have shown an increase in the thin-wall width compared to the width of weld tracks. The average difference between the thin-wall width and the weld track width was 18%. The largest differences in the average wall width and deposition width were found at Ls of 450 mm/s, reaching up to 30% (Figure 6a). At Ls of 500–750 mm/s, the measured difference was 15–17% (Figure 6b). It was also discovered that as Ls increases, the difference between the deposition width and thin-wall width decreases.

Aside from thin-wall thickness, another quantity examined was the minimum amount of energy required for the production process to achieve a quality fusion of the wall layers. Insufficient bonding of wall layers also occurred in the majority of samples with a surface energy density under 4.5 J/mm^2^. Samples produced with an energy density over 4.5 J/mm^2^ no longer contained defects between layers, with the exception of the Ls = 700 mm/s and Ls = 750 mm/s series. These no longer contained defects once the energy density reached 6 J/mm^2^. The test, therefore, shows that thin-walls can be produced using Lp and Ls combinations that did not achieve sufficient energy density during weld deposition testing. The thinnest wall without defects was produced at an Lp of 100 W and Ls of 500 mm/s, and its width was 169 μm.

### 3.2. Analysis of Surface Quality

The quality of the surface of the thin walls was determined based on images obtained using an optical profilometer. The acquired data clearly showed that a large number of powder particles partially fused to the surface of the sample. This significantly reduced the quality of the surface. Large sized particles led to locally high roughness, which resulted in a local loss of data on the surface typology. As a result, it was necessary to perform local iterative mapping of the surface, using Vision64 software. The recorded data were then used to compile charts (Figure 7) showing the dependence of the surface quality on the process parameters. The trend of change in the surface quality was affected by high surface roughness. Despite that, there was a noticeable increase in the surface roughness, with a growing surface energy value. The data also show the influence of Lp and Ls on the resulting surface quality. Changing Ls values appears to result in a smoother change in surface quality than changing Lp values. The lowest surface quality of 34.2 μm (Ra) was created at an Lp of 100 W and an Ls of 500 mm/s. The equation describing the development of surface quality, depending on the surface energy density (2), has a coefficient of determination of 97.68%.
*Y* = 25.8767 + 4.0362*x* − 0.1181*x*^2^(2)

### 3.3. Evaluation of Microhardness

Selected weld depositions were measured for microhardness according to Vickers method, at a load of 0.01 kg (0.098). Measurements were taken in two series. One was taken in the axis of the weld track, perpendicular to the base plate; the other was parallel to the surface at a distance of 0.075 mm below the sample surface (Figure 8a). Since microhardness values did not vary significantly within individual weld depositions, the resulting value was calculated as an average from all measurements in the weld deposition area. The resulting microhardness values were plotted on a chart and compared with the surface energy density and Lp values used to produce them (Figure 8b). Lp and surface energy density were chosen for closer examination since they have the largest effect on weld deposition shape.

### 3.4. Analysis of Local Chemical Composition

Etched weld depositions were examined using scanning electron microscopy. Local chemical composition of the material was measured at selected points using EDX point analysis. The microstructure in the weld deposition area is, in all cases, composed of coarse dendrites of a solid solution. It was possible to locally observe fine-grained eutectic systems, based on Mg-Y alloy WE43 [42,43], at the edges of the weld depositions. Further phase separation was not observed in the weld deposition area, nor at the interface between the substrate and the deposition area. Weld depositions created at Lp = 325 W, Ls = 650 mm/s, including details of the microstructure of its left portion and demarcation of areas, analysed via EDX, are shown in Figure 9. The detailed in Figure 9b shows the typical morphology of the lamellar eutectic. The results of the chemical analysis are shown in Table 2. The weld deposition area carried a small amount of Y and Nd (and also local incidence of small amounts of Si), corresponding to typical admixture elements found in the WE43 alloy powder used.

The appearance of weld deposition, created by parameter combination of Lp = 350 W, Ls = 550 mm/s, is documented in Figure 10. The image, captured via reflection electron imaging, shows a more pronounced separation of additive elements and general mixing of the deposited metal (see the lighter areas on Figure 10a), compared to the previous case. A detailed image of the weld deposition edge (upper left portion of the deposition), once again shows individual phases of the eutectic system (the difference in appearance, when compared to Figure 9b, is caused by a lower degree of etching of the metallographic sample). The upper right portion of the deposition (see Figure 10c) shows a fused powder particle consisting of a homogeneous phase of magnesium alloy that did not mix its melted surroundings. EDX microanalysis of the chemical composition (Table 3), in the area of the eutectic system, detected the presence of Y, Nd and Zr (yellow points). No zirconium was detected in the weld deposition area outside of the eutectic system. EDX mapping of the upper right area of the deposition (see Figure 10d–f) shows increased Y content in the subsurface area. This supports the hypothesis of the presence of a fused particle, whose surface is made up of yttrium oxide as a result of high temperature and diffusion processes. To check of hypothesis, EDX analysis of the chemical composition of the substrate was performed. As expected, only Mg and O were detected (see points 7 and 8 in Table 3).

## 4. Discussion

### 4.1. Sample Shape and Continuity

The purpose of the test was to find energy thresholds at which a continuous weld track can be created. Division of the weld tracks into individual areas (Figure 3), which were then used to compile a process map, was performed based on a comparison of their shape, continuity and surface energy density. A similar division is commonly used for weld tracks in other materials [42,43,44].

The lower energy threshold for creating continuous weld tracks was 5.5 J/mm^2^. Using lower energy density values resulted in an unstable melt pool. The melt pool’s collapse, due to surface tension of the liquid, resulted in the creation of separate melt pools that did not merge together into a continuous weld track, as is mentioned in literature [43,45] Low energy density also resulted in poor joining of the deposited material with the base plate. There were also local occurrences of the balling effect in low-energy areas. Increasing the energy density to over 5.5 J/mm^2^ made the weld tracks more continuous, up to an energy density of 15 J/mm^2^, where the weld track depth reached 300 μm or greater. The high energy density also led to evaporation of the powder material in the form of black fume as was present by [46,47,48]. The fume caused local defocusing of the laser beam, which was then unable to weld the powder to the base plate. The end result was a collapse of the melt pool and discontinuity in the weld track. Evaporation of the material also led to keyhole porosity at the weld root [36]. Issues with heavy evaporation appeared in some weld depositions, even at 12 J/mm^2^ and onward. The discovered energy density limits for individual areas (Figure 3) partially overlap. The reason for the ambiguous thresholds is the different effects that Lp and Ls have on the production process, which the calculated surface energy density cannot fully account for.

Lp and Ls values of 150 W and 200 mm/s, respectively, resulted in unstable weld tracks. The melt pool splashed out and weld depositions reached both very high and very low penetration depths. Even small changes in Lp by 25 W and Ls by 50 mm/s led to major differences in the weld deposition shape. One of the reasons was a higher percentage change in energy when changing Lp and Ls values in area A, compared to other areas. Creating a more detailed description of area A would require a reduction in the increases in Lp and Ls, so that energy density changes more smoothly. Another reason was the low melting point and evaporation point of the magnesium matrix in contrast with the high melting point of the oxides (MgO, Y2O3, etc.) on the surface of the powder and their high reflectance value. Penetration of the oxide layer can only be achieved through a higher laser power. A similar principle is used in aluminium alloys [49]. When comparing the results of the weld deposition test of aluminium alloy AlSi10Mg [44], it is evident that the processing of magnesium alloys requires higher energy densities. Even studies focusing on other magnesium alloys confirm that a higher laser power leads to a high relative density of volumetric samples [26,32,50]. However, increasing the energy density also leads to a rapid release of Mg vapours, which then oxidise to create MgO [50]. The oxide is visible during production in the form of a black mist. The difference in the amount of MgO released by changing the energy density by ca. 20 J/mm^2^ was even visible to the naked eye. A similar effect was found when processing other magnesium alloys via SLM [26,50,51].

When comparing the width of weld depositions and thin walls, it is evident that Lp has a stronger influence on the resulting sample width than Ls does. This finding is in line with behaviour observed in other magnesium alloys [23,28]. A gradual reduction in the energy density creates a melt pool with progressively smaller dimensions, which also leads to a decrease in the depth and width of the weld tracks. However, it its apparent, at first sight, that the thin walls are wider than the weld tracks. This shift was likely caused by a difference in the size of the melt pool. The melt pool size is dependent on the process parameters and temperature gradient [52], which decreases as the distance from the surface of the base plate grows. Lp has a greater effect on the size of the melt pool than Ls, a behaviour also found in other alloys [53]. A higher conduction of heat, away from the melt pool during welding, reduces the melt pool width and, thus, also the weld track width. A similar difference in the widths of the thin-wall and weld deposition samples has already been noted by Krauss and Zaeh [54]. The difference in the width between the thin walls and weld deposits was 18% on average.

However, the trend in the sample width was the same for both walls and weld depositions. It was still clear that increasing the energy density increases the sample width. Increasing the Ls from 450 mm/s to 750 mm/s reduced the offset between the weld depositions and thin walls. Increasing the Ls resulted in smaller melt pools and thus, lower weld depositions and thin-wall widths. Hyer et al. [36] provided similar weld deposit results for different spot size (70 μm) and Lt (40 μm). The reason for a higher drop in the thin-wall width, compared to weld depositions, was different heat distribution. The surrounding powder on the surface of the thin walls served as an insulant, meaning the highest flow of thermal energy led from the melt pool down towards the base plate. The weld depositions penetrated 3–6 layers in the thin-wall samples, which led to the melting and resolidification of a large portion of the thin-wall. Each melting also melted the surrounding powder, which increased the height of the thin-wall. This repeated remelting did not occur in the weld deposition samples. However, increasing the speed resulted in a decrease in the weld deposition depth and, thus, a decrease in the depth of the area where the thin walls were repeatedly melted. Therefore, reductions in Ls had a greater effect on the width of the thin walls than on the width of the weld depositions.

A cumulation of thermal energy in the thin walls also shifted the threshold for creating stable weld tracks from 5.5 J/mm^2^ to 4.5 J/mm^2^.

### 4.2. Analysis of Surface Quality

The aim of the test was to determine the dependence of changes in the surface quality of thin-wall samples on process parameters. The surface of the thin walls was highly rugged for all Lp and Ls combinations. The causes of this roughness were two-fold: firstly, the occurrence of a balling effect, which appeared at an energy density range of 2–4 J/mm^2^, and, secondly, the fusing of powder particles onto the sample surface.

The roughness of the surface, due to the balling effect, was most noticeable on weld depositions created at an Lp of 100 W (Figure 7). The occurrence of the balling effect had several causes. At low Lp values, it was caused by poor melt wettability, as previously described [45,55]. With increased energy density, the surface tension of the melt decreased, and the weld tracks became more continuous, which helped smooth out their surface [43,56]. The same behaviour was also detected in the thin-wall samples.

As energy grew further, the balling effect became more prominent as the melt reacted with the oxygen released from the magnesium melt during the production process. This reaction cannot be fully prevented, even by the presence of an inert atmosphere in the production chamber of the 3D printer [22,23,24,42,57]. The reason is the erosion of oxide film and the melting of the solid solution by laser impact, which releases bound oxygen. The oxygen will then quickly start reacting with the magnesium melt, which leads to the formation of small spherical particles on the sample surface. This was also observed by Salehi et al. [58]. The particles on the surface of the samples primarily consisted of Mg, Y and Nd oxides and compounds, as shown by the analysis of the local chemical composition and Liu et al. [59]. The growing energy density also made it easier for powder particles to cling to the surface of the sample, which, then, had further negative effects on surface quality. The dependence of surface quality on the energy density is described by Equation (2). A similar dependence was also found in titanium alloy Ti6Al4V [39].

The surface quality of the parts affects their corrosion resistance. Since magnesium alloys are highly susceptible to corrosion, improving the surface quality is one of the potential ways to combat this issue. The discovered dependencies can serve as groundwork for future research into the matter.

### 4.3. Microhardness

Measurement of the progression of microhardness across the weld deposition area (along the weld deposition axis and perpendicular to it) revealed no deviations in individual values. The hardness of the material in the weld deposition area was always constant. Weld depositions created, using different process parameters, showed slight differences in average hardness (35 to 45 HV0.01). However, no direct connection between microhardness and individual process parameters was established. It is apparent that the microhardness value has a direct correlation only with dendrite (grain) size in the weld deposition area, which is a function of the combination of all process parameters.

### 4.4. Microstructural Analysis and Local Microanalysis of Chemical Composition

Microstructural analysis revealed an occasional occurrence of eutectic in the border surface areas of weld depositions, which was in accordance with the research conducted in [36]. The formation of a eutectic system, based on Mg-Y (or Mg-Y-Zr), is a consequence of diffusion processes during heat exposure of the weld deposition. The presence of eutectic systems in the weld deposition area can be considered undesirable. Low-melting structural components are often a cause of increased internal tension in the material and in the formation of shrinkage cavities.

No phase transformations in the weld deposition area or on the interface between deposited metal and substrate were detected. The weld deposition area consists of a solid solution that is more or less homogeneous depending on process parameters.

Local EDX microanalysis revealed no significant diffusion between the area with the deposition metal (WE43 alloy powder) and the base substrate (wrought Mg 99.9%).

## 5. Conclusions

The main purpose of the article is to clarify the behaviour of weld tracks made from WE43 magnesium alloy with various combinations of process parameters. The acquired and interpreted results are to serve as a foundation for researching strategies on how to process WE43 magnesium alloy. The main results of this paper are as follows:-Stable and continuous weld depositions can be achieved within an energy density range of 5.5–12 J/mm^2^.-The layering of weld deposits on thin walls led to an average increase of 15–17% in the width of the melt pool.-The layering of the weld deposits led to reducing the required energy density for a stable weld track to 4.5 J/mm^2^.-An equation for estimating surface roughness based on process parameters has been created.

The change of the laser power and scan speed did not affect the microhardness and the chemical composition of the weld deposit.

## Figures and Tables

**Figure 1 materials-13-02623-f001:**
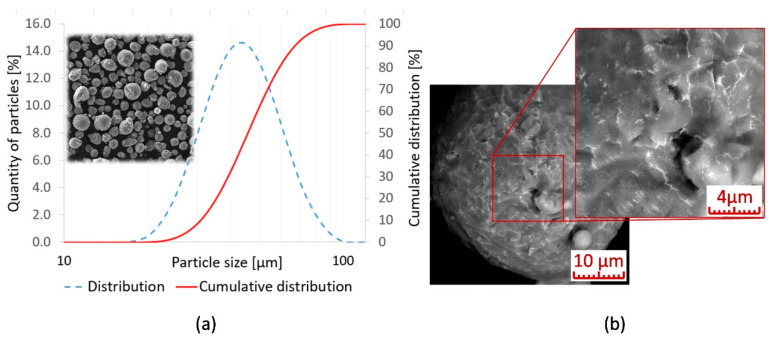
(**a**) Particle size distribution in the powder; (**b**) powder particle surface covered by oxides.

**Figure 2 materials-13-02623-f002:**
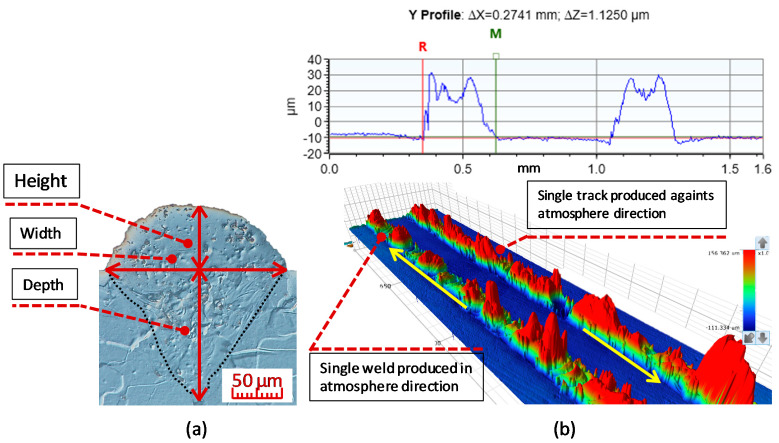
(**a**) Metallographic section of a weld track with highlighted fusion edges of base material and the marking of monitored dimensions; (LM-DIC, Nital); (**b**) measurement of weld track width.

**Figure 3 materials-13-02623-f003:**
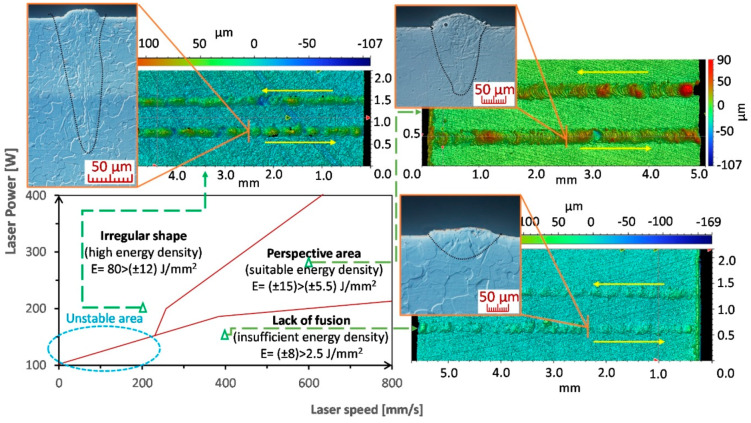
Experimental process map of weld tracks based on changes in surface energy density.

**Figure 4 materials-13-02623-f004:**
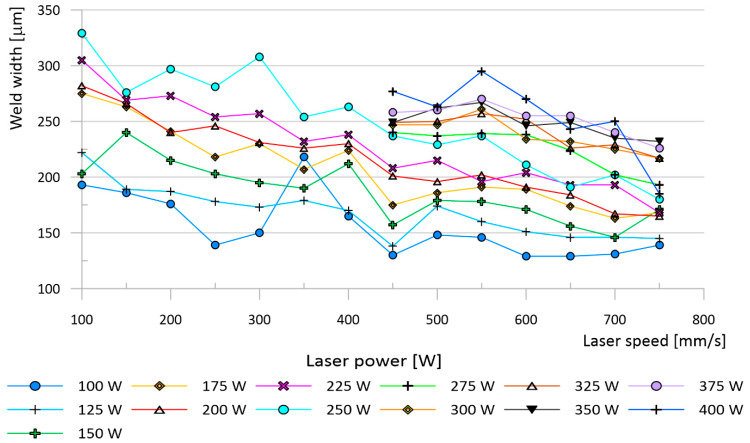
Variation in weld track width depending on scan speed and laser power.

**Figure 5 materials-13-02623-f005:**
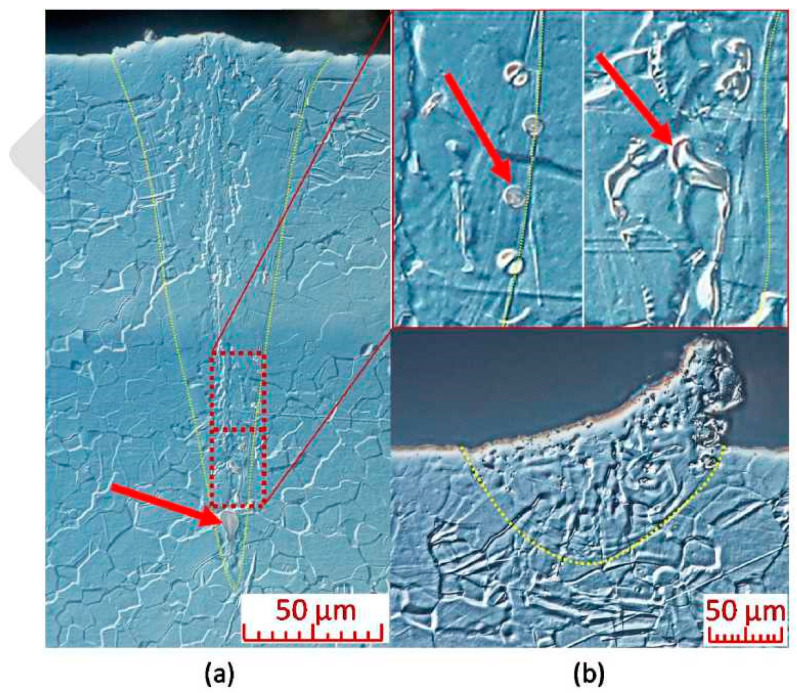
(**a**) Key-hole porosity in the weld root and porosity at the weld track edge (Lp 200 W, Ls 100 mm/s); (**b**) Unstable melt pool, resulting in splash and subsequent hardening of the melt (Lp 125 W, Ls 150 mm/s); (LM-DIC, Nital). Lp is laser power, Ls is scan speed.

**Figure 6 materials-13-02623-f006:**
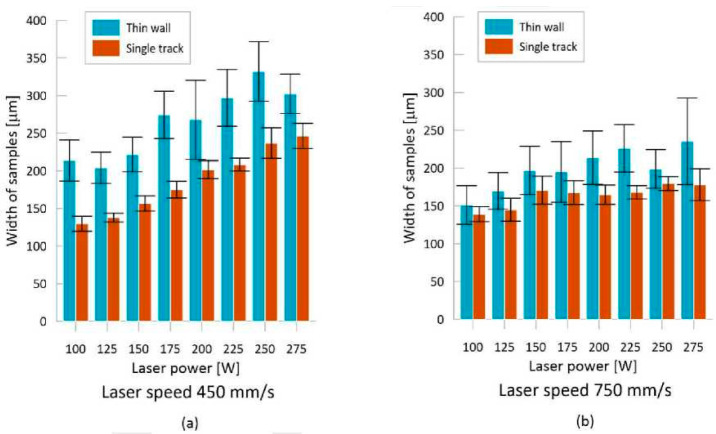
Quantification of the difference in the width of thin walls and weld tracks: (**a**) Ls 450 mm/s; (**b**) Ls 750 mm/s.

**Figure 7 materials-13-02623-f007:**
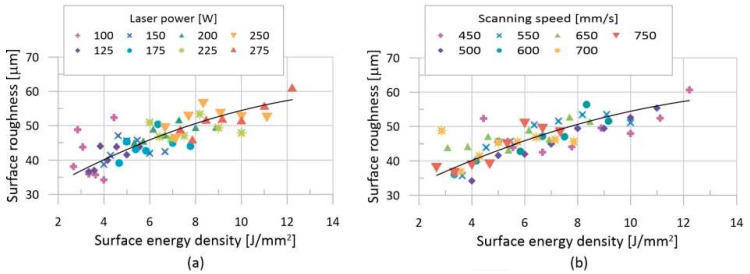
Arithmetic deviation in surface quality depending on the surface energy: (**a**) divided by Lp; (**b**) divided by Ls.

**Figure 8 materials-13-02623-f008:**
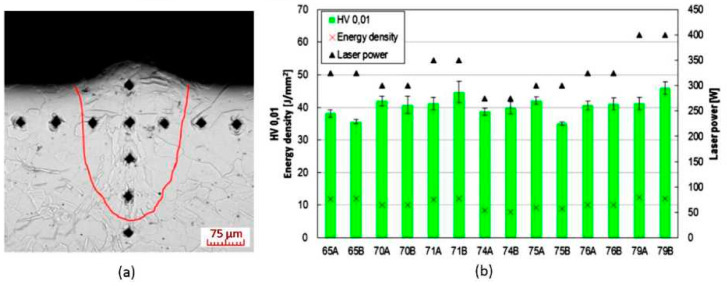
(**a**) Measurement of weld deposition microhardness (Lp = 350 W, Ls = 550 mm/s) with highlighted fusion penetration area; (LM, Nital); (**b**) average hardness values of selected weld tracks.

**Figure 9 materials-13-02623-f009:**
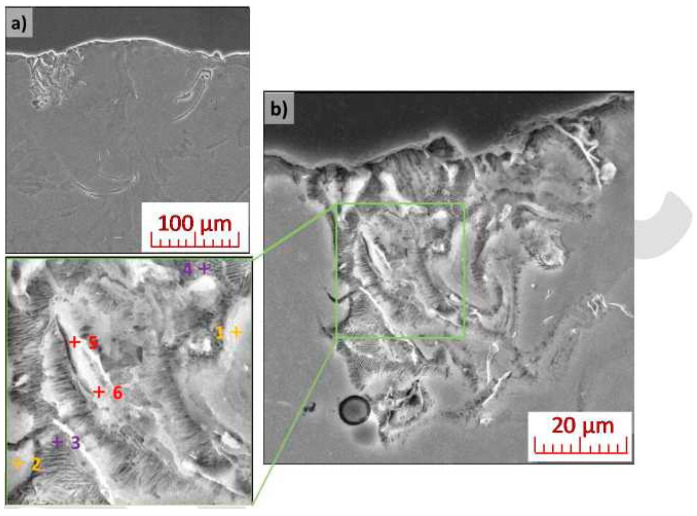
Weld deposition structure (Lp = 325 W, Ls = 650 mm/s): (**a**) general view; (**b**) mixed left portion of the weld deposition with sectioned detail and demarcation of the target points for EDX analysis.

**Figure 10 materials-13-02623-f010:**
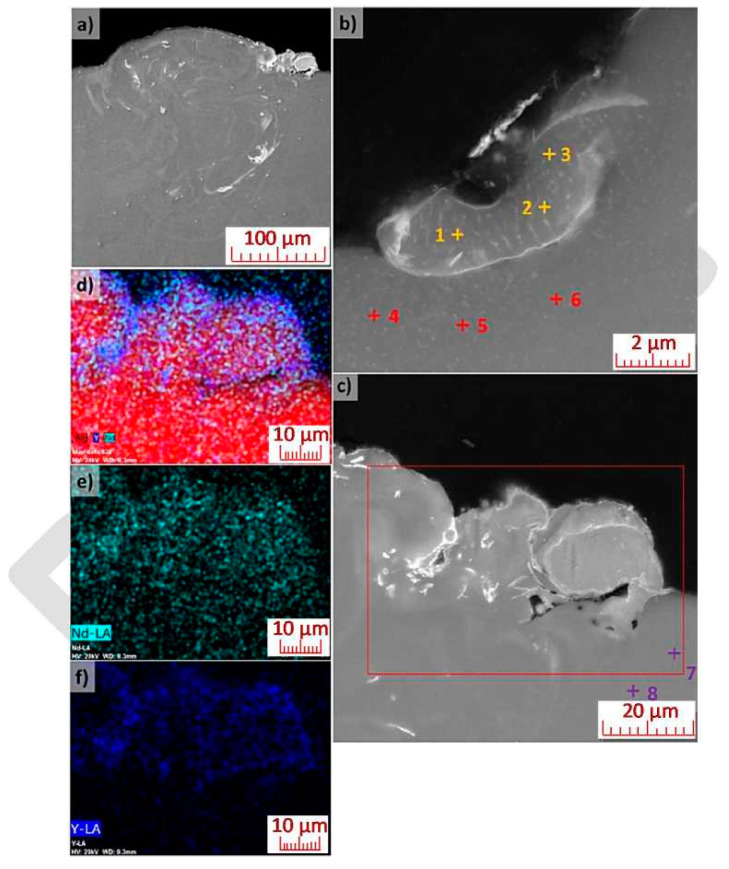
Weld deposition appearance (Lp = 350 W, Ls = 550 mm/s): (**a**) general view; (**b**) detail of mixing of the left portion of the weld deposition with marked target points for analysis; (**c**) detail of mixing of the right portion of the weld deposition with marked target points for analysis; (SEM-BSE, Nital); (**d**)–(**f**) EDX mapping (Mg, Y, Nd).

**Table 1 materials-13-02623-t001:** Chemical composition of powder used and normative thresholds (wt%).

The Mark of a Sample	Y	Zr	Nd	Si	Cu
**ASTM B93/B93M-06**	3.7–4.3	0.3–1.0	2.0–2.5	Max 0.01	Max 0.01
**WE43 powder**	3.96	0.56	2.30	<0.01	<0.01

**Table 2 materials-13-02623-t002:** EDX point analyses of local chemical composition of areas highlighted in Figure 9 (at.%).

Area	Mg	O	Si	Y	Nd
1	97.3	2.4	ND	0.2	0.1
2	96.3	3.0	0.4	0.2	0.1
3	97.9	0.4	ND	0.4	1.3
4	94.3	5.0	ND	0.4	0.3
5	93.6	5.1	0.7	0.5	0.1
6	96.1	2.5	0.6	0.7	0.1

**Table 3 materials-13-02623-t003:** Local chemical composition of areas marked in Figure 10b,c (at.%).

Area	Mg	O	Y	Nd	Zr
1	92.2	6.5	0.8	0.4	0.1
2	93.2	6.0	0.4	0.3	0.1
3	90.9	8.2	0.5	0.3	0.1
4	96.4	3.1	0.3	0.2	ND
5	97.4	2.4	0.1	0.1	ND
6	96.4	3.4	0.1	0.1	ND
7	97.2	2.8	ND	ND	ND
8	97.2	2.8	ND	ND	ND

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
