# Peer review of "Effect of Laser Parameters on Processing of Biodegradable Magnesium Alloy WE43 via Selective Laser Melting Method"

_materials, 2020, doi:10.3390/ma13112623_

Round 1

Reviewer 1 Report

1) Why the layer (Lt) is 50 μm?

2) Equat. 2.1, explanation of parameters.

3) Microhardness, for which time?

4) Why the weld width decreases with rising laser speed?

5) Fig. 8a, how the surface hill is created? 

Author Response

English language and style are fine/minor spell check required

The text was subsequently corrected by a native speaker.

  1. Question

Why the layer (Lt) is 50 μm?

  1. Answer

The layer thickness is selected according to the particle size distribution of the powder. The standard range is 20 - 100 µm. The benefit of low layer thickness is an increase in the relative density of the part produced (1). On the other hand, increasing the layer thickness lead to reduce the amount of oxygen released during the 3D printing of magnesium alloys (2). Paradoxically, therefore, it would theoretically be possible to improve the part quality by increasing the layer thickness. However, due to the large depth of the weld deposit, the layer thickness does not play a decisive role in the quality of the part produced in comparison with laser power, scanning speed and hatch distance (3). The biggest benefit of layer thickness increasing is the decrease of build time.

A lot of researchers used the layer thickness of 30 mm for processing magnesium alloy WE43 (4–6). It is possible because they print specimen on the AconityMINI machine. This machine was developed for the printing of difficult processable material as Mg alloy. As a result, they are less confronted with problems such as smoke vapor cloud and aggressive material oxidation during printing (7). Tandon et. al. (8) used a very similar 3D printer as our team for WE43 printing and with the layer thickness 50 mm reached relative good results. We decided to use the layer thickness 50 mm for these reasons.

  1. Question

Equat. 2.1, explanation of parameters.

  1. Answer

Thank you for your attention. An explanation of the parameters was added to the manuscript text below equation 2.1. There was also a mistake in equation 2.1. We exchanged minus for multiplication in the denominator.

  1. Question

Microhardness, for which time?

  1. Answer

The loading time for the hardness test was 15s. In accordance with current standard EN ISO 6507-1, the loading time is not noted in the hardness designation, if the test load time is from 10 to 15s. For this reason, time is not included in the hardness measurement methodology.

  1. Question

Why the weld width decreases with rising laser speed?

  1. Answer

The range of the laser speed has a direct influence on the melt pool size (3,9). A less laser speed leads to energy concentration and an increase of melt pool size. The melt pool grows in length but not width by increasing the laser speed because the energy is spread over time over a larger (longer) area. Another interpretation is that the laser speed increase reduces the time that laser affects a certain area. Therefore the melt pool can’t grow up so much in size and also the weld track size doesn't grow up logically too. This behaviour is similar for different alloys, for magnesium alloy AZ91 it is shown here (3).

  1. Question

Fig. 8a, how the surface hill is created?

  1. Answer

We have to apologize because we are not sure what you mean by the surface hill. The weld track cross-section is shown in figure 8a with points after microhardness measurement as is mean in material and methods chapter (2.4. Measurement of Microhardness and Local Chemical Composition). The weld track is created by the melting of the metal powder by the laser beam as you can see here (10–14).

  1. Savalani MM, Hao L, Dickens PM, Zhang Y, Tanner KE, Harris RA. The effects and interactions of fabrication parameters on the properties of selective laser sintered hydroxyapatite polyamide composite biomaterials. Rapid Prototyp J [Internet]. 2012;18(1):16–27. Available from: http://www.emeraldinsight.com/doi/10.1108/13552541211193467
  2. Savalani MM, Pizarro JM. Effect of preheat and layer thickness on selective laser melting (SLM) of magnesium. Rapid Prototyp J [Internet]. 2016;22(1):115–22. Available from: http://www.emeraldinsight.com/doi/10.1108/RPJ-07-2013-0076
  3. Schmid D, Renza J, Zaeh MF, Glasschroeder J. Process influences on laser-beam melting of the magnesium alloy AZ91. Phys Procedia. 2016;83:927–36.
  4. Zumdick NA, Jauer L, Kersting LC, Kutz TN, Schleifenbaum JH, Zander D. Additive manufactured WE43 magnesium: A comparative study of the microstructure and mechanical properties with those of powder extruded and as-cast WE43. Mater Charact [Internet]. 2019;147(August 2018):384–97. Available from: https://doi.org/10.1016/j.matchar.2018.11.011
  5. Gangireddy S, Gwalani B, Liu K, Faierson EJ, Mishra RS. Microstructure and mechanical behavior of an additive manufactured (AM) WE43-Mg alloy. Addit Manuf [Internet]. 2019;26(September 2018):53–64. Available from: https://doi.org/10.1016/j.addma.2018.12.015
  6. Bär F, Berger L, Jauer L, Kurtuldu G, Schäublin R, Schleifenbaum JH, et al. Laser additive manufacturing of biodegradable magnesium alloy WE43: A detailed microstructure analysis. Acta Biomater [Internet]. 2019;(xxxx). Available from: https://doi.org/10.1016/j.actbio.2019.05.056
  7. Jauer L, Meiners W, Vervoort S, Gayer C, Zumdick NA, Zander D. Selective laser melting of magnesium alloys. In: World PM 2016 Congress and Exhibition. European Powder Metallurgy Association (EPMA); 2016.
  8. Tandon R, Palmer T, Gieseke M, Noelke C. Additive Manufacturing of Magnesium Alloy Powders: Investigations Into Process Development Using Elektron®MAP+43 Via Laser Powder Bed Fusion and Directed Energy Deposition. Euro PM2016. 2016;91:4–9.
  9. Louvis E, Fox P, Sutcliffe CJ. Selective laser melting of aluminium components. J Mater Process Technol [Internet]. 2011;211(2):275–84. Available from: http://dx.doi.org/10.1016/j.jmatprotec.2010.09.019
  10. Aboulkhair NT, Maskery I, Tuck C, Ashcroft I, Everitt NM. On the formation of AlSi10Mg single tracks and layers in selective laser melting: Microstructure and nano-mechanical properties. J Mater Process Technol. 2016;230:88–98.
  11. GUO Y, JIA L, KONG B, WANG N, ZHANG H. Single track and single layer formation in selective laser melting of niobium solid solution alloy. Chinese J Aeronaut [Internet]. 2018;31(4):860–6. Available from: https://doi.org/10.1016/j.cja.2017.08.019
  12. Ng CC, Savalani MM, Man HC, Gibson I. Layer manufacturing of magnesium and its alloy structures for future applications. Virtual Phys Prototyp. 2010;5(1):13–9.
  13. Chung Ng C, Savalani M, Chung Man H. Fabrication of magnesium using selective laser melting technique. Rapid Prototyp J [Internet]. 2011;17(6):479–90. Available from: http://www.emeraldinsight.com/doi/10.1108/13552541111184206
  14. Khairallah SA, Anderson AT, Rubenchik A, King WE. Laser powder-bed fusion additive manufacturing: Physics of complex melt flow and formation mechanisms of pores, spatter, and denudation zones. Acta Mater [Internet]. 2016;108:36–45. Available from: http://dx.doi.org/10.1016/j.actamat.2016.02.014

Reviewer 2 Report

The aim of paper “Effect of laser parameters on processing of 3 biodegradable magnesium alloy WE43 via selective 4 laser melting method” is to clarify the effects of laser power 75 and scan speed on the production process, to find the dependence of size and continuity of weld 76 tracks on energy density and to perform a basic analysis of surface quality, depending on the process 77 parameters. The main idea of the paper is not clear in the manuscript. There are very few comparisons with other works in the bibliography. It is a merely descriptive article. I don not consider the interest to the readers

Author Response

  1. Question

The aim of paper “Effect of laser parameters on processing of 3 biodegradable magnesium alloy WE43 via selective 4 laser melting method” is to clarify the effects of laser power 75 and scan speed on the production process, to find the dependence of size and continuity of weld 76 tracks on energy density and to perform a basic analysis of surface quality, depending on the process 77 parameters. The main idea of the paper is not clear in the manuscript. There are very few comparisons with other works in the bibliography. It is a merely descriptive article. I don not consider the interest to the readers.

  1. Answer

Thank you for your feedback. We admit that the manuscript contained some mistakes which were fixed during the revision process. We rewrote the abstract, conclusion, and expanded the introduction and discussion section. Also, the text was subsequently corrected by a native speaker. We believe that the text and the aim of the manuscript are now clearer. We see the greatest benefit of the manuscript in laying the foundation for further developing the 3D printing strategy which will lead to an improved surface roughness of the structure grids and manufacturing parts. The surface roughness is one of the greatest problems by additive manufactured parts as is described by literature [1–3].

  1. Zhang, W. neng; Wang, L. zhi; Feng, Z. xue; Chen, Y. ming Research progress on selective laser melting (SLM) of magnesium alloys: A review. Optik (Stuttg). 2020, 207, 163842.
  2. Gockel, J.; Sheridan, L.; Koerper, B.; Whip, B. The influence of additive manufacturing processing parameters on surface roughness and fatigue life. Int. J. Fatigue 2019.
  3. Li, Y.; Zhou, J.; Pavanram, P.; Leeflang, M.A.; Fockaert, L.I.; Pouran, B.; Tümer, N.; Schröder, K.U.; Mol, J.M.C.; Weinans, H.; et al. Additively manufactured biodegradable porous magnesium. Acta Biomater. 2018, 67, 378–392.

Reviewer 3 Report

Dear Authors,

the paper is well done, however, one thing should be verified: the load and notation concerning the Vickers hardness at page 9. The normal notation HV0,01 (Fig 8b) means the load was 0,01kgf and the method was micro-hardness measurement. In line 238 the load is given as 10kg (shoud be kG or kgf?). This looks wrong, what more such load means a macro-Vickers method (load up to 30kgf). Please, make controll.

Author Response

English language and style are fine/minor spell check required

The text was subsequently corrected by a native speaker.

  1. Question

Dear Authors,

the paper is well done, however, one thing should be verified: the load and notation concerning the Vickers hardness at page 9. The normal notation HV0,01 (Fig 8b) means the load was 0,01kgf and the method was micro-hardness measurement. In line 238 the load is given as 10kg (shoud be kG or kgf?). This looks wrong, what more such load means a macro-Vickers method (load up to 30kgf). Please, make control

  1. Answer

Thank you for your typo warning. At the declared hardness HV0,01, the load is really 0.01kgf (0.098N).

Reviewer 4 Report

The authors did excellent research work and well written manuscript. The objectives,  results and discussions are clear and going well with the subject of this work. I recommend the acceptance of this manuscript in its present form with minor revisions of writing styles.

Author Response

Moderate English changes required

The text was subsequently corrected by a native speaker.

  1. Question

The authors did excellent research work and well written manuscript. The objectives,  results and discussions are clear and going well with the subject of this work. I recommend the acceptance of this manuscript in its present form with minor revisions of writing styles..

  1. Answer

We have slightly expanded the discussion, changed the abstract and made minor changes to the text on the base of the other reviews. We also let the text go through the family spokesperson and we believe that the text is now easier to understand. Thank you very much for your review and we wish you much success in the future.

Round 2

Reviewer 1 Report

After the corrections, the manuscriptnow  can be published.

Reviewer 2 Report

The paper "Effect of laser parameters on processing of 3 biodegradable magnesium alloy WE43 via selective 4 laser melting method" has been carefully reviewed and rewritten and I think now it may be suitable for publication.

Reviewer 3 Report

The following version looks OK.